# A Frame-to-Frame Scan Matching Algorithm for 2D Lidar Based on Attention

Shan Huang [1,2] and Hong-Zhong Huang [1,2,*]

1 School of Mechanical and Electrical Engineering, University of Electronic Science and Technology of China, Chengdu 611731, China; huangshan@std.uestc.edu.cn
2 Center for System Reliability and Safety, University of Electronic Science and Technology of China, Chengdu 611731, China
* Correspondence: hzhuang@uestc.edu.cn; Tel.: +86-28-6181252

**Abstract:** The frame-to-frame scan matching algorithm is the most basic robot localization and mapping module and has a huge impact on the accuracy of localization and mapping tasks. To achieve high-precision localization and mapping, we propose a 2D lidar frame-to-frame scanning matching algorithm based on an attention mechanism called ASM (Attention-based Scan Matching). Inspired by human navigation, we use a heuristic attention selection mechanism that only considers the areas covered by the robot's attention while ignoring other areas when performing frame-to-frame scan matching tasks to achieve a similar performance as landmark-based localization. The selected landmark is not switched to another one before it becomes invisible; thus, the ASM cannot accumulate errors during the life cycle of a landmark, and the errors will only increase when the landmark switches. Ideally, the errors accumulate every time the robot moves the distance of the lidar sensing range, so the ASM algorithm can achieve high matching accuracy. On the other hand, the number of involved data during scan matching applications is small compared to the total number of data due to the attention mechanism; as a result, the ASM algorithm has high computational efficiency. In order to prove the effectiveness of the ASM algorithm, we conducted experiments on four datasets. The experimental results show that compared to current methods, ASM can achieve higher matching accuracy and speed.

**Keywords:** 2D lidar; scan matching; attention; robotics



## 1. Introduction

Frame-to-frame scan matching is the process of obtaining the relative pose between two frames whose visual fields overlap with one another. Frame-to-frame scan matching is a basic module for robot localization and mapping and has a great impact on the accuracy of localization and mapping. High-precision frame-to-frame scan matching can significantly improve the loop detection accuracy while reducing the computational burden of loop detection. Therefore, frame-to-frame scan matching plays an important role in robot state estimation. A high-precision frame-to-frame scan matching algorithm is critical to improve the autonomous navigation capabilities of robots.

Frame-to-frame matching algorithms can be divided into two categories based on the type of sensor used: laser frame-to-frame matching algorithms and visual frame-to-frame matching algorithms. Compared to visual sensors, laser lidar has anti-interference capabilities, and its performance is not affected by light. Moreover, the research results of this paper are mainly applied to indoor service robots, so 2D lidar sensors were adopted for scan matching.

Due to the importance of the frame-to-frame scan matching algorithm, it has attracted much research attention, and research has resulted in many milestone findings. The most classic frame-to-frame scan matching algorithm is the Iterative Closest Point (ICP) algorithm proposed by Besl et al. [1]. ICP associated each lidar point in the current frame

with the lidar points in the reference frame to construct constraints, allowing the relative pose be obtained with the Horn method [2], and the relative pose was then applied in the current frame. The ICP continues to repeat this process until convergence. A large number of variants [3] have been proposed based on ICP. Andrea [4] proposed Point-to-Line ICP (PL-ICP), which overcomes the shortcomings of low-resolution lidar data and has quadratic convergence properties, achieving fewer iterations and higher accuracy, and this algorithm performed well in a structured environment. Similar to the PL-ICP, Segal et al. [5] proposed a probabilistic version of the ICP: GICP (Generalized ICP), which uses Gaussian distribution to model the lidar sensor data and assigns weights according to the normal vector. This algorithm can be considered to be a variant of Point-to-Plane ICP [6]. In order to speed up the scan matching ability of the GICP, Koide [7] proposed the use of voxels to accelerate nearest neighbor searching. Serafin [8] introduced normal vector and curvature information into the ICP. He first filtered wrong matches with the normal vectors and curvature and then added the normal vector alignment error term into the error function to obtain a more accurate angle estimation. Deschaud [9] used the Implicit Moving Least Square (IMLS) method to model the surface in the point cloud and proposed the IMLS-ICP, building an accurate surface model so that the scan matching accuracy was greatly improved. Liu et al. [10] proposed a precise point set registration method that introduced correntropy measurements to weaken the influence of noise and introduced an adaptive feature fusion algorithm that was distribution specific. Yin et al. [11] proposed a novel probabilistic variant of the iterative closest point algorithm that leverages both local geometrical information and the global noise characteristics. Liao et al. [12] proposed new ICP variants that used fuzzy clusters to represent scans with a broad convergence basin and with good noise robustness. Moreover, they also developed a branch-and-bound-based global optimization scheme that was able to minimize the metrics globally regardless of the initialization technique. Li et al. [13] have carried out a number of studies to determine impact factors such as the overlap ratio, angle difference, distance difference, and noise to come up with a global solution to the ICP algorithm. Liu et al. [14] proposed a global point set registration method that decouples the optimization of translation and rotation. Additionally, the global translation parameter was obtained first using a fast branch-and-bound method, and then the optimal rotation parameter was calculated using the global translation parameter.

In addition to the ICP algorithm and its variants, there are different methods for frame-to-frame scan matching. Biber et al. [15] proposed the Normal Distribution Transform (NDT) algorithm, which assumes that the environmental structure has local continuity and constructs multiple local Gaussian distributions to represent the overall geometric structure of the environment and achieves registration by minimizing the distance between the lidar point to the normal distribution. The NDT method is widely used in robot mapping and localization tasks [16]. Joe et al. [17] proposed a sonar scan matching method based on NDT that combined two different sonars to reduce the drift errors. Bouraine et al. [18] proposed a scan matching method based on NDT and particle swarm optimization that could determine the global optimal solution with 70 particles. To overcome the influence of local extrema on the matching results, Olson [19] proposed the Correlative Scan Match (CSM) method, which divides the search area into grids and then enumerates the poses corresponding to each grid to obtain the robot's position. The CSM method avoids the influence of local extreme values through enumeration. Ren et al. [20] proposed an improved CSM method that analyzes environmental degradation with a covariance matrix, which greatly improved the robustness of the algorithm in complex environments. Kohlbrecher et al. [21] proposed the construction of a local grid map and then generated a continuous likelihood map with Lagrangian interpolation before using the Gauss–Newton method to register the algorithm. Ali et al. [22] proposed a novel end-to-end trainable deep neural network for rigid point set registration: RPSRNet, which used a novel $2^D$-tree representation for the input point sets and hierarchical deep feature embedding in the neural network. Zhang et al. [23] proposed the lidar odometry and mapping (LOAM)

method, which utilizes edge features and planar features in the lidar scans. LOAM is very suitable for structured environments. To improve the accuracy, Zhang et al. [24] proposed fusing lidar and vision to construct a more accurate odometry. Shan et al. [25] proposed a lightweight and ground-optimized lidar odometry and mapping (Lego-LOAM) algorithm that leverages the presence of a ground plane during its segmentation and optimization steps and that utilizes two-step optimization to solve different components of the transform matrix.

All of the methods above use two consecutive frames for scan matching, and the pose is integrated frame by frame, so the matching error also accumulates frame by frame. Although strategies have been adopted to minimize the matching error between frames, the increase in the error frame by frame still limits their matching accuracy. Moreover, these methods use most of the laser points for scan matching, resulting in a low matching efficiency. Although we can integrate data sampling methods [26,27] to improve the computational efficiency, this would reduce the matching accuracy.

To overcome the shortcomings of the above methods, we introduce an attention mechanism and propose a scan matching method to achieve both high accuracy and computational efficiency. This paper introduces an attention mechanism into the frame-to-frame scan matching algorithm. Different from the above methods, the proposed method only accumulates errors when the attention area is switched. Additionally, the proposed method only uses the lidar data covered by the attention area, so it has very high computational efficiency.

First, we selected a frame as a reference frame and chose an object as a landmark that was far enough away. The size of the landmark must be larger than the given threshold, and the landmark must provide enough information to obtain the relative pose; secondly, we chose the odometer to determine the initial pose of the selected frame and to determine the relative pose between the current frame and the landmark, extracting the corresponding data. Thirdly, the relative pose between the current frame and the reference frame was obtained based on the corresponding data. Finally, if the selected landmark reached the edge of the visual field of the current frame, then a new landmark was selected from the current frame as the reference landmark for the subsequent frame, and the current frame became the new reference frame. Landmark selection is part of the attention construction process, and once the attention has been constructed, the scan matching algorithm only considers the area covered by the attention and ignores other areas. As a result, the number of required calculations is greatly reduced. Since most current lidars have a measurement range of 20 m, the robot can move 20 m without accumulating errors using the ASM method in the ideal conditions. The main contributions of this work are as follows:

(1) This paper proposes the concept of an attention mechanism. Additionally, we introduce this mechanism into the frame-to-frame scan matching algorithm, significantly improving the accuracy and computational efficiency of the scan matching algorithm.
(2) This paper proposes attention area selection, attention area update, and attention area scan matching methods that successfully integrate the attention mechanism into the scan matching algorithm.
(3) The proposed attention-based scan matching algorithm is evaluated on multiple real-world datasets, and it outperforms the current state-of-the-art methods in terms of accuracy and efficiency.

## 2. System Overview

The ASM algorithm can be divided into three modules: the attention area selection module, the pose solving module, and the results verification and key frame selection module. The overall system framework is shown in Figure 1.

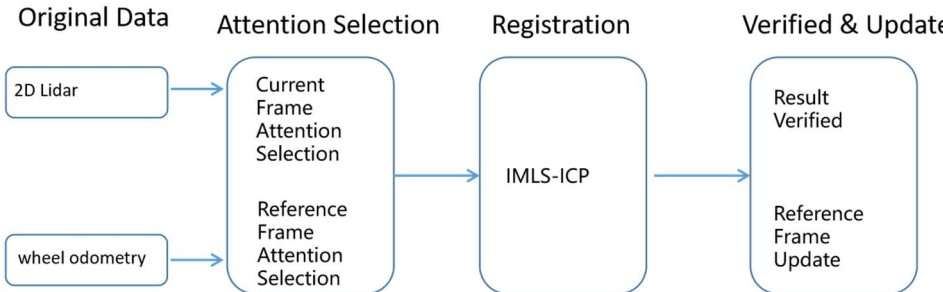

**Figure 1.** System overview.

The ASM algorithm takes 2D lidar data and wheel odometry as its input. The wheel odometry is used to determine the robot's movement distance relative to the previous keyframe, and when the distance exceeds the threshold, a new key frame is generated, and the algorithm enters the matching process. First, the attention area selection module selects the area that the matching algorithm needs to pay attention to while ignoring all other areas; secondly, the algorithm uses the laser data obtained from the attention area to match the reference frame and to obtain the pose of the current key frame; Finally, the coincidence degree of the attention area is evaluated according to the matching results, and if the coincidence degree exceeds the threshold, then the matching pose is used as the final pose; otherwise, the odometer pose is used as the final pose.

The pseudo code of the proposed method is shown in Algorithm 1. The three modules are further described in Sections 3–5.

---

**Algorithm 1:** The Proposed Algorithm.

---

**Input:** reference frame lidar data $ld_{rf}$, reference frame pose $P_{rf}$, reference frame attention area $AA_{rf}$, wheel odometry pose $P_o$, now frame lidar data $ld_{nf}$
$\delta$—distance threshold for keyframe insertion.
**if** $|P_{rf} - P_o| > \delta$ **then**
    Calculation the attention area of now frame $AA_{nf}$;
    Calculation the pose of now frame $P_{nf}$ using the lidar data covered by attention area;
    **if** *LegalPoseCheck($P_{nf}$) == false* **then**
        **return**;
    **end**
    **if** *isReferenceFrameNeedUpdate() == false* **then**
        **return**;
    **end**
    $ld_{rf} = ld_{nf}$;
    $P_{rf} = P_{nf}$;
    $AA_{rf} = AA_{nf}$;
**end**

---

## 3. Attention Area Selection Module

Selecting the attention area is critical to the performance of the matching algorithm. The selected attention area mainly operates the reference frame and the current frame. In the reference frame, attention area selection needs to determine which landmark in an environment should be used for localization. In the current frame, attention area selection refers to determining the matching data from the area presented in the current frame and in the reference frame based on the attention area of the reference frame and the initial pose of the current frame.

### 3.1. Reference Frame Attention Area Selection

In the reference frame, attention area selection is equivalent to a human selecting a landmark when he or she is walking. Once a landmark is selected, no errors can be accumulated until the landmark completely disappears. Therefore, landmark selection greatly

impacts the matching results. When conducting multiple experiments, it is understood that better performance is achieved when the following three conditions of the attention area are satisfied:

(1) The attention area must be away far enough from the original pose of the reference frame.
(2) The selected attention area must ensure that there is enough information that can be obtained from the pose of the current frame.
(3) There must be a sufficient number of laser points in the selected attention area.

The distance in condition (1) can be considered as the drift-free running distance of the ASM algorithm. Obviously, when the other two conditions are met, the longer the distance, and the higher the scan matching accuracy. Since 2D laser data has angle constraints naturally, the constraint in condition (2) refers to the translation constraint in the *X*-axis direction and the *Y*-axis direction, and the selected attention area must provide sufficient constraints in both the *X*-axis and *Y*-axis directions. Condition (3) ensures the stability of the solution through the use of enough laser points. This is because when the number of laser points is too small, then the solution that is obtained is unstable even if there are sufficient constraints.

Obviously, condition (3) is easy to implement, and it only requires the number of lidar points in the attention area to be counted. However, condition (1) can be satisfied by selecting the area the farther away from the areas that meet the condition (2) and condition (3), and condition (2) requires additional processing.

When a certain key frame is determined as the reference frame, the normal vector of each lidar point in the reference frame is calculated. For a certain point $p_i$, the surrounding point cloud obeys the Gaussian distribution, and the covariance matrix of the Gaussian distribution is obtained. For 2D data, the covariance matrix is two-dimensional, the eigenvector corresponding to its large eigenvalue represents the direction vector of the point, and the eigenvector corresponding to the smallest eigenvalue represents the normal vector of the point. In order to obtain the normal vector of $p_i$, the covariance matrix of the Gaussian distribution is calculated using Equation (1):

$$
\begin{aligned}
u_i &= \frac{1}{|N_i|} \sum_{p_j \in N_i} p_j \\
\Sigma_i &= \frac{1}{|N_i|} \sum_{p_j \in N_i} (p_j - u_i)^T (p_j - u_i),
\end{aligned}
\tag{1}
$$

where $u_i$ represents the mean value of the Gaussian distribution. $\sum_i$ represents the covariance matrix of the Gaussian distribution, and $N_i$ represents the set of points near $p_i$. The calculation of the normal vector requires the covariance matrix to perform eigenvalue decomposition:

$$
\Sigma_i = \begin{bmatrix} v_1 & v_2 \end{bmatrix} \begin{bmatrix} \lambda_1 & 0 \\ 0 & \lambda_2 \end{bmatrix} \begin{bmatrix} v_1^T \\ v_2^T \end{bmatrix},
\tag{2}
$$

where $\lambda_1$ and $\lambda_2$ represent the eigenvalues of the covariance matrix to be $\lambda_1 > \lambda_2$. $v_1$ and $v_2$ are the eigenvectors corresponding to the eigenvalues. Moreover, the normal vector expression of point pi is shown in Equation (3).

$$
n_i = v_2
\tag{3}
$$

When the normal vector is known, the translation constraint of point $p_i$ in the *X*-axis and *Y*-axis directions needs to be computed. The translation of the constraints of point $p_i$ on the *X*-axis and the translation on the *Y*-axis are shown in Equations (4) and (5), respectively:

$$
c_{ix} = |n_i \cdot X_i|,
\tag{4}
$$

$$
c_{iy} = |n_i \cdot Y_i|,
\tag{5}
$$

where $X_i$ and $Y_i$ represent the direction vector of the *X*-axis and the direction vector of the *Y*-axis, respectively. When the normal vector of a point is parallel to the coordinate axis, the constraint on that axis reaches its maximum value; when the normal vector of the point is perpendicular to the coordinate axis, the constraint on this axis is the smallest. Therefore, it is feasible to evaluate the contribution of a point to the translation constraint using the inner product of the vector with the direction vector of the coordinate axis.

Based on the above description, Figure 2 depicts the flow diagram and the schematic diagram of the selected attention area of the reference area.

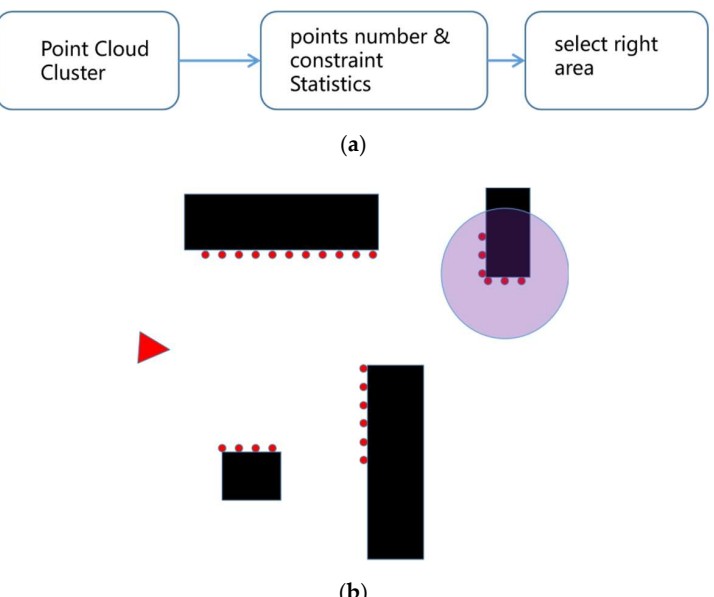

**Figure 2.** Selection of the attention area in the reference frame. (**a**) Flow diagram; (**b**) schematic diagram. The red triangle is the robot, the red dots are the lidar data, and the purple circle is the attention area.

First, the point clouds of the reference frame are clustered by the region growing segmentation method [28], and the clusters with too few points are removed; secondly, the number of points in each cluster and the average contribution of the point clouds to the *X*-axis constraints and *Y*-axis constraint is calculated; finally, the cluster that does not meet the threshold is removed, and the cluster that is the farthest from the origin of the reference frame among all of the clusters that meet the threshold is selected as the new landmark. The area corresponding to the landmark is the selected attention area. In Figure 2b, the red triangle represents the position of the robot, the red dots represent the lidar data, the black squares represent an obstacle in the environment, and the purple circle represents the selected attention area. It is clear that the selected attention area has sufficient constraints and is far enough away from the robot's position.

### 3.2. Current Frame Attention Area Selection

In the current frame, attention area selection determines the data that should be used to match the reference frame to the current frame. In this case, where the attention area of the reference frame and the initial pose $(x_0, y_0, \theta_0)$ of the current frame are known, Equation (6) is used to convert the data from the current frame into a world coordinate system through the initial pose:

$$
\begin{aligned}
p_w &= T(x_0, y_0, \theta_0) \cdot p_c = T \cdot p_c \\
T &= \begin{bmatrix} \cos(\theta_0) & -\sin(\theta_0) & x_0 \\ \sin(\theta_0) & \cos(\theta_0) & y_0 \\ 0 & 0 & 1 \end{bmatrix},
\end{aligned}
\tag{6}
$$

where $p_c$ represents the point cloud in the current frame coordinate system, and $p_w$ represents the point cloud in the world coordinate system.

The attention area of the reference frame is expanded. If the point in the current frame is located in the expansion layer of the attention area of the reference frame, then the point needs to be matched with the landmark of the reference frame, as shown in Figure 3.

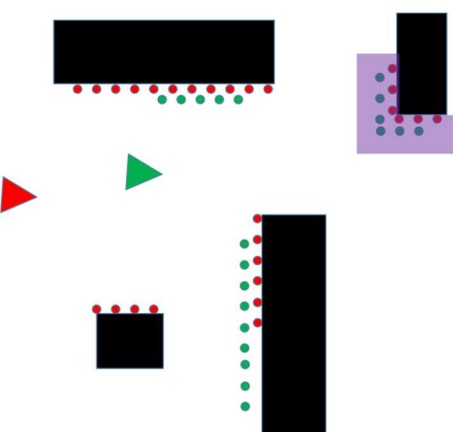

**Figure 3.** Data association in the attention area of the current frame. Red is the reference frame, green is the current frame, and the purple area is the attention area.

In Figure 3, the red triangle and the red dots represent the reference frame and its laser data, the green triangle and the green dots represent the current frame and the corresponding laser data, and the purple-red area represents the expansion layer of the reference frame's attention area. The initial pose of the current frame is determined by wheel odometry, but due to the limited accuracy of wheel odometry, the lidar data of the current frame does not coincide with the reference frame. This error is compensated for through the expansion layer. All of the lidar points in the purple-red area in the current frame are used to match the landmarks of the reference frame, and the set of these points is the attention area of the current frame.

### 4. Pose Matching Module

After the attention area has been selected, two-point cloud sets from the attention area of the reference frame and from the attention area of the current frame are available. The pose matching module needs to calculate the relative poses of these two-point cloud sets. There are many ways to solve point cloud registration problems. To minimize the impact of sensor noise, a scan-to-model method [9] that uses the implicit moving least squares method (IMLS) was adopted to model the plane of the point cloud obtained from the reference frame. The distance from the laser point $p$ in the current frame to the plane is shown in Equation (7) [29]:

$$
\begin{aligned}
d(p) &= \frac{\sum\limits_{p_i \in S} W_i(p)((p-p_i) \cdot n_i)}{\sum\limits_{p_i \in S} W_i(p)} \\
W_i(p) &= e^{\frac{-\|p-p_i\|^2}{\sigma^2}},
\end{aligned}
\tag{7}
$$

where $S$ represents the point cloud of the attention area in the reference frame, $n_i$ represents the normal vector of the corresponding laser point, $\sigma$ is the artificially set attenuation factor, and $W$ is an exponential function, and it decays quickly with distance and with the consideration of the reference frame points where the distance $p$ exceeds $3\sigma$ is not necessary. Equation (7) implicitly expresses the hidden surface in the reference frame. It assumes that the sampling noise of the point cloud to the surface obeys Gaussian distribution, so the number and distribution of the sampling points on both sides of the surface are the same.

Moreover, the points on the surface are substituted into Equation (7), and the obtained distance value d is theoretically equal to 0; that is to say, all of the points satisfying $d = 0$ constitute a hidden surface.

Based on Equation (7), the objective function of scan-to-model matching method is obtained and is presented as follows:

$$f = \sum_{p_i \in C} |d(T \cdot p_i)|^2,$$ (8)

where $C$ represents all of the point clouds in the attention area of the current frame, $T$ represents the transformation matrix corresponding to the robot pose, and its expression was shown earlier in Equation (6).

Equation (8) represent an exponential function that is a nonlinear least square problem. In order to solve the registration problem more effectively, a point $q_i$ on the surface is obtained as a matching point for each point $p_i$ in the current frame, and the objective function of the matching method is shown in Equation (9).

$$f = \sum_{p_i \in C} |T \cdot p_i - q_i|^2$$ (9)

Since the initial pose of the current frame is determined by wheel odometry, the error rate can be controlled within a small range. When the angle error is small, a small angle approximation can be introduced [30], and $T$ can be expressed as

$$T = \begin{bmatrix} 1 & -\theta & x \\ \theta & 1 & y \\ 0 & 0 & 1 \end{bmatrix}$$ (10)

By introducing the approximation shown in Equation (10), Equation (9) can be simplified as follows:

$$\begin{aligned} T \cdot p_i - q_i &= \begin{bmatrix} 1 & -\theta & x \\ \theta & 1 & y \\ 0 & 0 & 1 \end{bmatrix} \begin{bmatrix} p_{ix} \\ p_{iy} \\ 1 \end{bmatrix} - \begin{bmatrix} q_{ix} \\ q_{iy} \\ 1 \end{bmatrix} \\ &= \begin{bmatrix} p_{ix} - p_{iy}\theta + x - q_{ix} \\ p_{ix}\theta + p_{iy} + y - q_{iy} \\ 0 \end{bmatrix} \end{aligned}$$ (11)

Obviously, Equation (11) is a linear equation system, and the independent variables can be extracted:

$$\begin{aligned} T \cdot p_i - q_i &= \begin{bmatrix} 1 & 0 & -p_{iy} \\ 0 & 1 & p_{ix} \end{bmatrix} \begin{bmatrix} x \\ y \\ \theta \end{bmatrix} - \begin{bmatrix} q_{ix} - p_{ix} \\ q_{iy} - p_{iy} \end{bmatrix} \\ &= A_i X - b_i \end{aligned}$$ (12)

If we substitute Equation (12) into Equation (9), then we can achieve

$$f = \sum_{p_i \in C} |A_i \cdot X - b_i|^2 = |AX - b|^2$$ (13)

Equation (13) is a linear least squares equation, where $A$ and $b$ can be expressed as follows:

$$A = \begin{bmatrix} 1 & 0 & -p_{0y} \\ 0 & 1 & p_{0x} \\ 1 & 0 & -p_{1y} \\ 0 & 1 & p_{1x} \\ \vdots & \vdots & \vdots \\ 1 & 0 & -p_{ny} \\ 0 & 1 & p_{nx} \end{bmatrix}, b = \begin{bmatrix} q_{0x} - p_{0x} \\ q_{0y} - p_{0y} \\ q_{1x} - p_{1x} \\ q_{1y} - p_{1y} \\ \vdots \\ q_{nx} - p_{nx} \\ q_{ny} - p_{ny} \end{bmatrix} \tag{14}$$

For the linear least squares problem described in Equation (13), matrix $A$ can be decomposed by means of SVD, and the singular vector corresponding to the minimum singular value of matrix $A$ is the solution of the corresponding problem.

## 5. Result Verification and Key-Frame Selection Module

Through the above two modules, the pose of the current frame can be determined. Next, the current frame needs to be verified to determine whether the pose of the current frame is accurate. Moreover, whether the reference frame needs to be updated according to the position of the reference frame landmark in the current frame needs to be determined. The purpose of verifying the matching result is to prevent a jump problem in the matching pose caused by insufficient constraints. Two methods can be used to verify the post, with the first method calculating the difference between the wheel odometry pose and the scan matching pose. The pose difference is defined in Equation (15). If the difference is less than a certain threshold, the scan matching pose is considered legal; otherwise, the scan matching pose is illegal.

$$err = T2V(T_{odom} \cdot T_{sm}^{-1}), \tag{15}$$

Here, $T_{odom}$ represents the transformation matrix corresponding to the odometry pose, $T_{sm}$ represents the transformation matrix corresponding to the scan matching pose, and $T2V$ () represents the function that extract the robot pose from the transformation matrix and is expressed as follows:

$$T = \begin{bmatrix} \cos\theta & -\sin\theta & x \\ \sin\theta & \cos\theta & y \\ 0 & 0 & 1 \end{bmatrix} \tag{16}$$
$$T2V(T) = (x, y, \theta)$$

The second method evaluates the correctness of the pose using the degree of overlap between the attention area of the current frame and the reference frame. Since the two areas represent the same object in the physical space, the degree of overlap can be large if the pose is correct. The degree of overlap in the two areas is defined as the reference area generating a distance field, each grid storing the distance to the nearest obstacle, and the current area being converted to the distance field. If the distance from the obstacle to the point is less than the threshold, then the point and the reference frame are considered to be overlapping. The number of overlapping points is counted, and the ratio of the number of overlapping points to the total number of points indicates the degree of overlap between the current area and the reference area. The degree of overlap ranges from 0 to 1. When the degree of overlap is greater than the threshold, then the pose is considered accurate; otherwise, the pose is considered inaccurate, and the odometry pose is used to replace the matching pose.

After the pose of the current frame is determined, whether the current frame needs to be updated to a new reference frame needs to be determined. If the current frame is to become the new reference frame, then it needs to meet two conditions: the distance of the current landmark from the origin of the current frame needs to be less than the threshold so that a suitable landmark can be selected in the current frame. If the current landmark is

close to the origin of the current frame, then a new landmark selection strategy needs to be undertaken. The new landmark selection method is shown in Section 3.1. If a landmark that meets the requirements is selected, then the current frame is used as the new reference frame, and the landmark corresponding to the current frame is used as the new landmark, as shown in Figure 4.

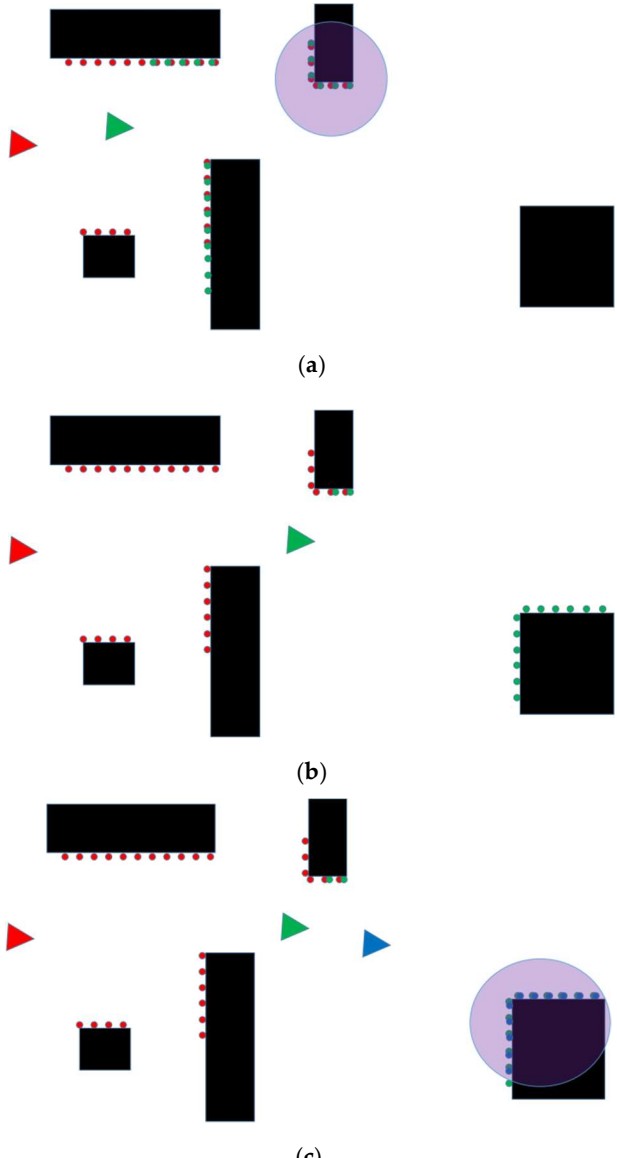

**Figure 4.** Reference frame and landmark update. (**a**) Initial attention area. Red is the reference frame, green is current frame, and the purple area is the current landmark; (**b**) attention area switching. The current landmark is close to the robot; (**c**) the attention area is switched, and blue is the newest frame, and the purple area is the new selected landmark.

In Figure 4a, the red triangle and red dots represent the reference frame and its laser data, the green triangle and green dots represent the current frame and the corresponding laser data, and the purple-red area represents the current landmark. As the robot continues to move forward, it comes to the position shown in Figure 4b, at which point the current landmark is very close to the robot's position and is about to leave the robot's field of view. Therefore, the reference frame and landmark need to be updated at this time. The current frame is used as the reference frame, and the landmark is selected to be the new reference landmark from the current frame. The new landmark is shown in Figure 4c, where the

green triangle represents the new reference frame, the blue triangle represents the current frame, and the purple-red area represents the new landmark.

## 6. Experiments

In order to verify the effectiveness of the ASM algorithm, experiments were conducted in four typical indoor scenarios, including offices, office buildings, libraries, and shopping malls. Cartographer [31] software was used to build 2D maps of the four test environments and are shown in Figure 5.

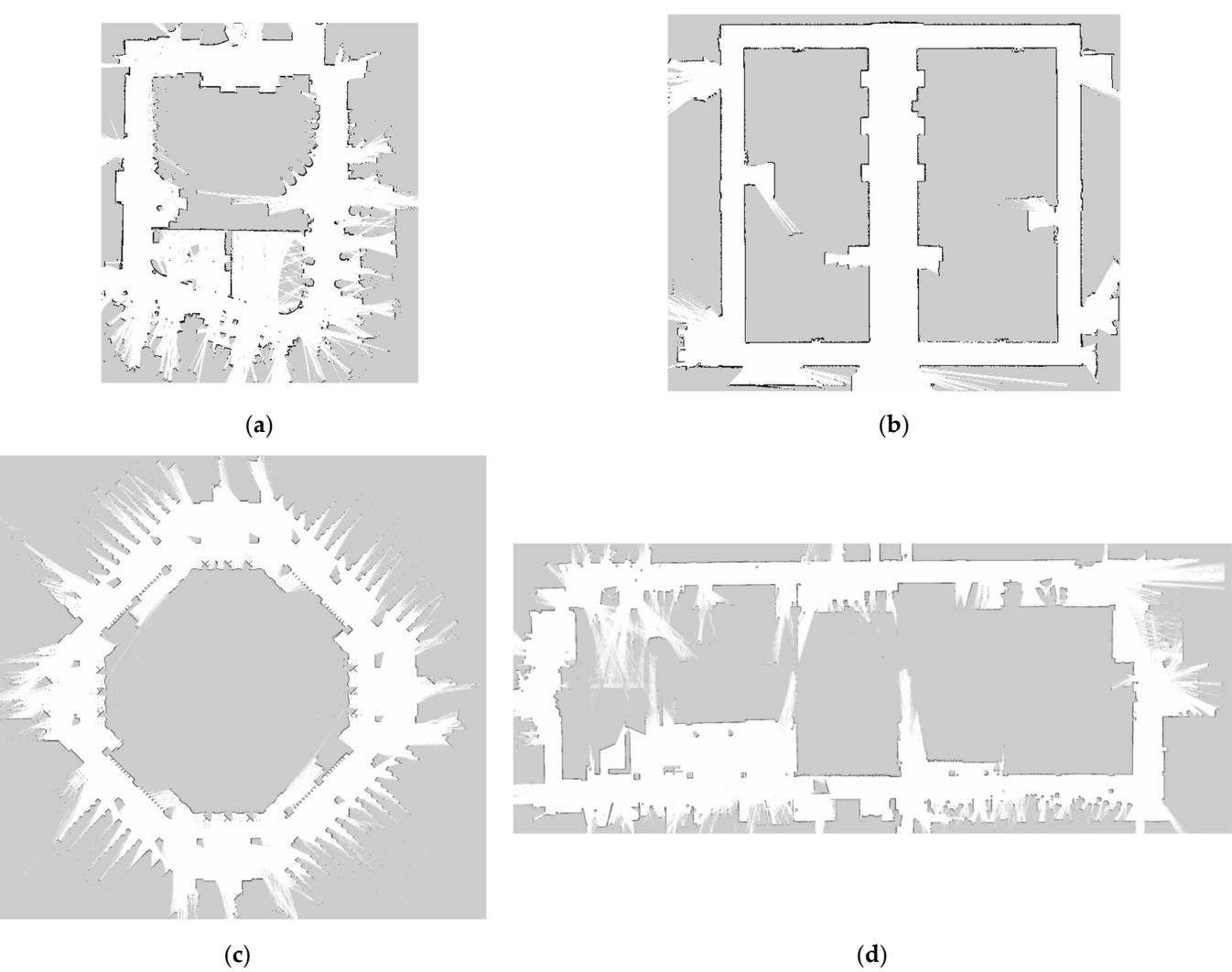

(**a**)

(**b**)

(**c**)

(**d**)

**Figure 5.** Maps of the test environment. (**a**) Office map; (**b**) office building map; (**c**) library map; (**d**) shopping mall map.

All four of the environments shown in Figure 5 have a back-shaped structure. The robot starts from the origin and makes a circle around the environment and then returns to the vicinity of its origin. These types of environments are very suitable for comparing accuracy with end-to-end error metrics.

In order to compare the presented method with existing methods, four representative methods: ICP [3], PL-ICP [4], iCSM [20], and IMLS-ICP [9] were selected. The ICP is implemented in the Point Cloud Library (PCL) [32], the PL-ICP method uses the author's own open-source code, the iCSM method uses the implementation in the cartographer [31] software with covariance matrix degradation analysis, and the IMLS-ICP method is not open source, and therefore, a 2D version was implemented according to the methodology

outlined in the paper in which it was published. The test platform used in this paper is shown in Figure 6.

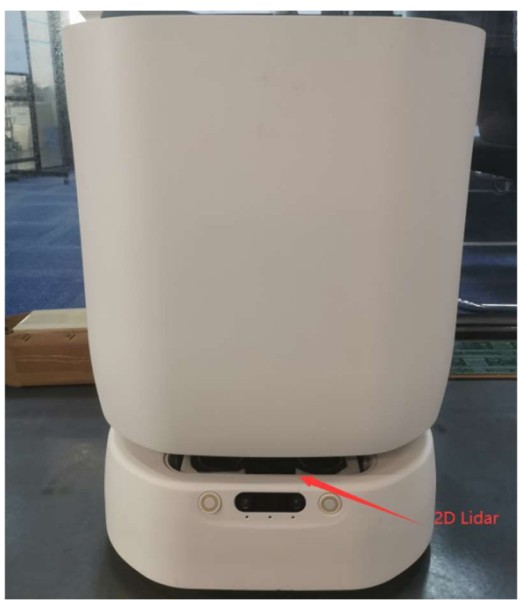

**Figure 6.** Test platform.

The test platform shown in Figure 6 is a differentially driven wheeled robot and is a prototype of the delivery robot created by Yuefan Innovation. The robot is equipped with a wheeled odometer and a low-cost 2D TOF lidar sensor. The update frequency of the wheel odometer is 100 Hz, the update frequency of the lidar data is 10 Hz, and the field of view is 250 degrees. The wheel odometer pose is used as the initial pose, and the scan matching algorithm is further optimized based on the initial pose. All programs run on an Rockchip ARM Core RK3399 process at 1.5 Ghz with 2 GB of memory.

Since it is impossible to obtain the true value of the robot's trajectory in these four datasets, the end-to-end error was used to compare the accuracy. When the robot starts from the origin position, moves around the environment, and then returns to the origin position, the spatial displacement of the same object in the environment is selected as the cumulative error of the frame-to-frame matching algorithm.

To ensure that the experimental results are not affected by different batches of experimental data. Before the start of the experiment, the robot was controlled to move in the four environments, and the data of the sensors carried by the robot were collected to form four datasets. All experiments are carried out on these datasets.

*6.1. Accuracy Test*

In order to evaluate the matching accuracy of the ASM algorithm and to compare the end-to-end error of different matching algorithms, the experiment was executed in four environments. The office maps constructed by the five matching algorithms are shown in Figure 7. The inconsistencies created by the matching algorithms are highlighted by the red circle. The green points represent lidar points. It can be seen in Figure 7 that the inconsistencies caused by the ASM algorithm are almost invisible. Additionally, the PP-ICP algorithm has the largest cumulative error, PL-ICP has the second-largest cumulative error, iCSM has the third-largest cumulative error, and IMLS-ICP has the fourth-largest cumulative error. The ASM algorithm has the smallest cumulative error. Additionally, the map built by the ASM algorithm is mostly consistent, which means that the cumulative error of the ASM algorithm on this data set is very small and outperforms other methods. The IMLS-ICP's cumulative error is slightly larger than ASM's cumulative error. The reason

for this is that the test environment is relatively small, allowing the IMLS-ICP to obtain better results.

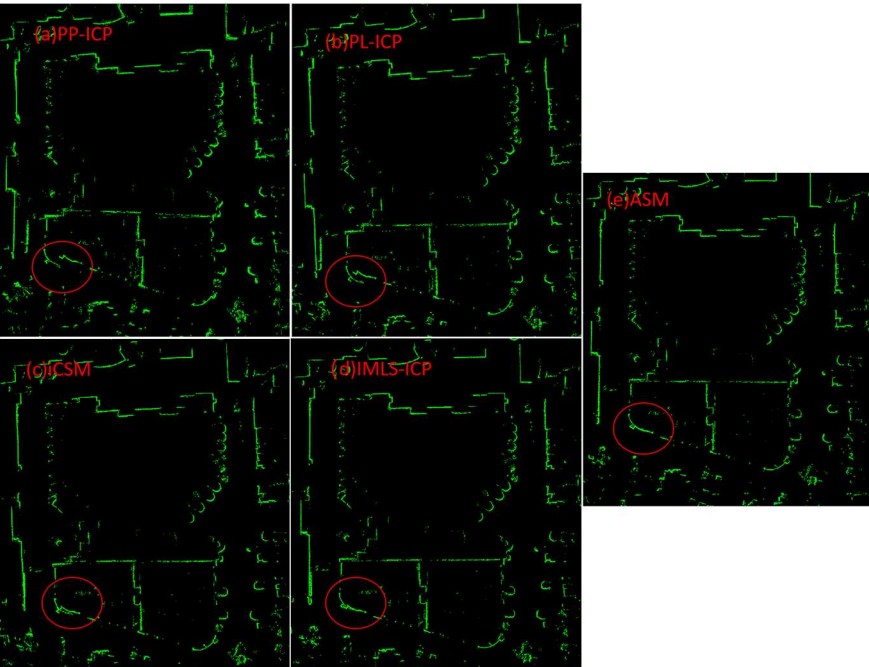

**Figure 7.** Office maps.

In addition to the office environment, we also tested the frame-to-frame scan matching methods in the library environment. The robot was still allowed to start from one point, move around the environment, and return to the same point. The robot built the map of the environment during the test based on the poses obtained by the scan matching algorithm. The maps of the library environment constructed by the five algorithms are shown in Figure 8. To display the differences more clearly, Only the inconsistencies in the library maps are shown. The red circle represents the inconsistencies in the maps caused by the cumulative error of the scan matching algorithm. It is clear that PP-ICP had the largest cumulative error followed by the PL-ICP algorithm, the CSM algorithm, and the IMLS-ICP algorithm. The ASM algorithm had the smallest cumulative error. The inconsistencies in the maps of the library environment built by the ASM algorithm are clearly greater than those in the office environment, as the robot had to travel over a much larger distance in the library. The ASM algorithm achieved similar results in the library environment as it did in the office environment, and its scan matching accuracy greatly outperforms that of the other four algorithms.

In addition, we also tested the inter-frame matching in the office building environment. The maps constructed by the five algorithms are shown in Figure 9. Once again, the red circle highlights the areas of inconsistency in the maps. Unlike the test results in the office and library environments, the PL-ICP algorithm, CSM algorithm, and IMLS-ICP algorithm show similar cumulative error rates. Because the office building environment is more structured compared to the office environment and the library environment, it is filled with many linear structures, so the different algorithms were able to achieve good results in this environment. Even in such a highly structured environment, the ASM algorithm was superior to the other four algorithms. There are no distinguishable inconsistencies in the map generated by the ASM algorithm. That is to say, in an office building environment, the ASM algorithm can build a globally consistent map while only relying on the scan matching pose, and the cumulative error the scan matching algorithm is very small.

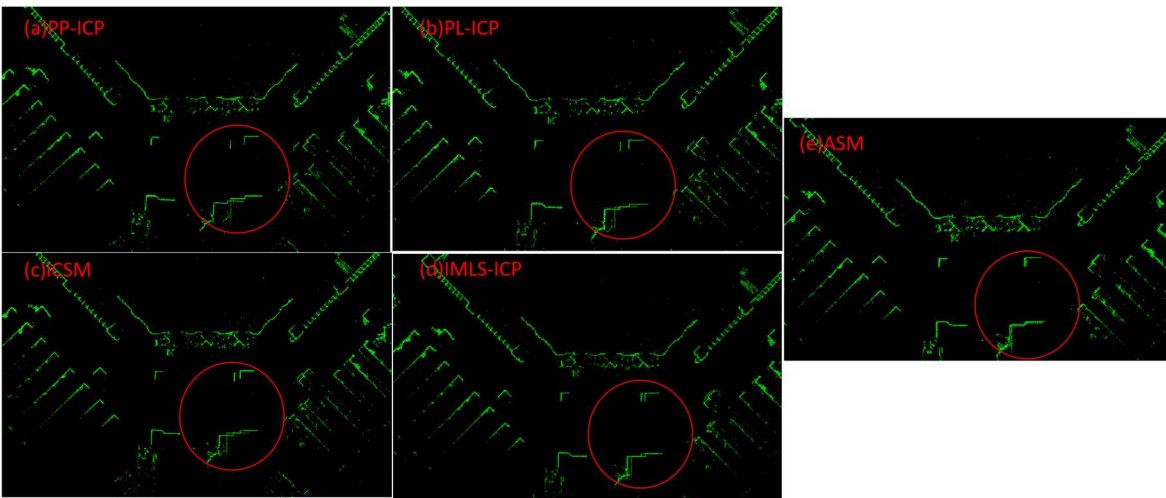

**Figure 8.** Library maps.

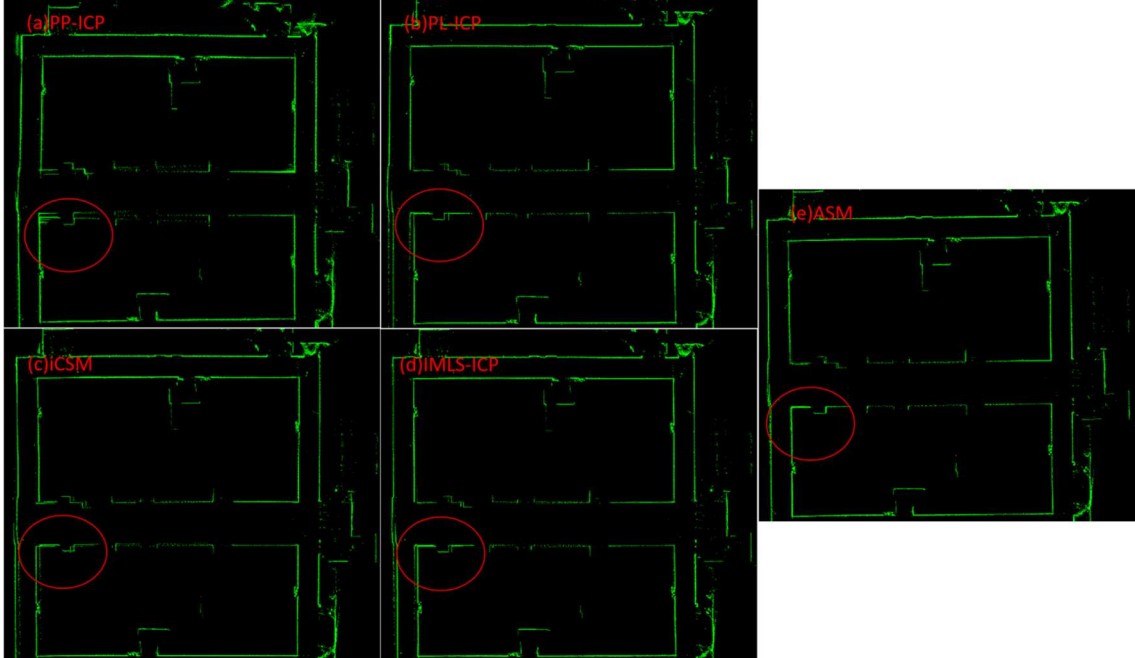

**Figure 9.** Office building maps.

To make a more fair comparison, we needed to prevent random factors from affecting the experimental results. To achieve this, we ran the five algorithms in the four test environments one hundred times. For each scan matching algorithm, we counted the data from these 100 experiments, calculated the cumulative error of each experiment, and used the average error of the 100 experiments as the final error to evaluate the matching accuracy of each scan matching algorithm. Table 1 shows the cumulative error statistics of each scan matching algorithm in the different environments.

**Table 1.** End-to-end error statistics table.

| Datasets | PP-ICP (cm) | PL-ICP (cm) | Icsm (cm) | IMLS-ICP (cm) | ASM (cm) |
|----------|-------------|-------------|-----------|---------------|----------|
| Office | 97.78 | 58.04 | 21.53 | 8.96 | **2.39** |
| Office building | 38.93 | 31.14 | 27.72 | 21.67 | **5.85** |
| Library | 120.03 | 72.05 | 52.67 | 27.43 | **11.71** |
| Shopping mall | 207.50 | 161.79 | 145.48 | 127.10 | **50.15** |

It can be seen from Table 1 that the ASM algorithm significantly outperforms the other four frame-to-frame matching algorithms in the four experimental environments. The reason for this is that the ASM method introduces an attention mechanism, so it is only possible for errors to accumulate when the attention area is switched. However, the frequency at which the attention area switches is very low. The robot only switches the attention area when the robot moves several tens of meters, so the cumulative error of the ASM algorithm is very small. The error of the other algorithms accumulate frame by frame, so the accumulated error will be larger. The superiority of the ASM algorithm is more prominent in the library and shopping mall environments. The reason for this is that the ASM algorithm can select a landmark from the reference frame that is far enough away from the origin of the reference frame in a large-scale environment. In large-scale environments, the longer lifetime of the reference landmarks results in the ASM algorithm having a longer drift-free movement distance. That is to say, the attention regions switch less frequently, and the error accumulation of the matching algorithm is smaller.

*6.2. Efficiency Test*

In addition to accuracy, computational efficiency is also very important for evaluating the scan matching algorithm. Similar to the accuracy test, the scan matching algorithm was repeatedly run on each dataset 100 times, and the average computation time required per each algorithm was used as the final evaluation standard. The statistical results are shown in Table 2.

**Table 2.** Computation time statistics table.

| Datasets | PP-ICP (ms) | PL-ICP (ms) | iCSM (ms) | IMLS-ICP (ms) | ASM (ms) |
|----------|-------------|-------------|-----------|---------------|----------|
| Office | 43.1 | 45.7 | 62.7 | 100.5 | **10.3** |
| Office building | 42.7 | 44.3 | 63.1 | 101.8 | **9.4** |
| Library | 44.3 | 45.0 | 63.5 | 102.3 | **9.71** |
| Shopping mall | 43.8 | 44.5 | 61.2 | 102.7 | **10.0** |

It can be seen from Table 2 that the time spent by each algorithm in different environments is similar, and this is because that the algorithm matching time is only related to the point cloud data and has nothing to do with the size of the environment. Although the office and office building environments are small in size, the number of point clouds in each frame of the laser data is similar, so the final matching efficiency is also similar. The most time-consuming algorithm is the IMLS-ICP algorithm because it needs to build a KD tree to find the nearest neighbors, which is time-consuming. The time spent by the PP-ICP and PL-ICP algorithms is basically the same, whereas the time spent by the ASM algorithm lower than that of the other algorithms. This is because the ASM algorithm only uses landmarks for matching tasks, and in most cases, the number of points in a landmark is less than 1/10 of the total number of points, so the matching speed can be greatly improved.

**7. Conclusions**

This paper introduced an attention mechanism and proposed a new frame-to-frame matching algorithm based on attention. The introduction of the attention mechanism

dramatically reduced the accumulated error rate and greatly improved the computational efficiency. An attention area selection and updated algorithm was proposed that successfully realized adaptive reference frame selection and the switching of different reference landmarks. To verify the effectiveness of the proposed algorithm, we conducted extensive experiments using real robots. Thanks to the attention mechanism, the accumulation error of the ASM method is very small, and the amount of the data involved in the calculation is also very small, which makes the ASM method is far superior to the other four methods discussed in this paper in terms of accuracy and computational efficiency in the four selected test environments. Additionally, the ASM method was able to achieve better results in a large-scale environment. The experimental results demonstrated that the robot could significantly improve navigation performance by simulating the way of human navigation, which can be regarded as a new exploration direction. In future work, semantic features will be introduced to define the attention area and to explore the attention area selection, update methods and improve the reliability in highly dynamic environments [33–36].

**Author Contributions:** Conceptualization, S.H. and H.-Z.H.; data curation, S.H.; methodology, S.H.; formal analysis, S.H. and H.-Z.H.; formal analysis, S.H. and H.-Z.H.; writing—original draft preparation, S.H. All authors have read and agreed to the published version of the manuscript.

**Funding:** This research was funded by [the National Key R&D Program of China] grant number [2017YFB1301300] and The APC was funded by [hong-zhong Huang].

**Institutional Review Board Statement:** Not applicable.

**Informed Consent Statement:** Not applicable.

**Acknowledgments:** Thanks to Qi Zeng for his help in the experiments.

**Conflicts of Interest:** The authors declare no conflict of interest.

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
