# Peer review of "A Frame-to-Frame Scan Matching Algorithm for 2D Lidar Based on Attention"

_applsci, doi:10.3390/app12094341_

Round 1
Reviewer 1 Report
This paper proposes a frame-to-frame scan matching algorithm for the 2D LIDAR method. In my opinion, there are still many parts that need to be improved.
1) In the introduction section, provide an explanation of the differences between the previous methods and the proposed method
2) Novelty and the proposed method are not clear
3) Figures 2 and 3 should be combined
4) Give text in Figure 4 to make it easier to understand
5) Figure 5 is not clear
6) In the results section, provide a comparison of results with previous methods, at least 5 published papers in 2017-2022
7) Provide comparisons with previous methods with performance metrics such as recall and accuracy
8) Reference at least the papers published in the past 5 years
Author Response
First of all, all authors would like to thank you for your valuable comments that are very important to improve our paper. We carefully reviewed these comments and gave detailed responses as follows:
- In the introduction section, provide an explanation of the differences between the previous methods and the proposed methodï¼›
Response: As suggested, we provide an explanation of the differences between the previous methods and the proposed method.
- Novelty and the proposed method are not clear..
Response: As suggested, we highlight the contribution and the novelty in the introduction.
- Figures 2 and 3 should be combined.
Response: As suggested, Figures 2 and 3 are combined.
- Give text in Figure 4 to make it easier to understand.
Response: As suggested, We have given a more detailed description to make it easier to understand.
- Figure 5 is not clear.
Response: As suggested, As suggested, We have given a more detailed description to make it easier to understand.
- In the results section, provide a comparison of results with previous methods, at least 5 published papers in 2017-2022.
Response: As suggested, We conduct more experiments and present the comparison results between our method and the classical method as well as the current state-of-the-art methods.
- Provide comparisons with previous methods with performance metrics such as recall and accuracy.
Response: As suggested, we provide comparisons with previous method with the accuracy and the efficiency in Table 1 and Table 2.
- Reference at least the papers published in the past 5 years.
Response: As suggested, We have added many recent reference papers and the number of reference papers has over 30.
Reviewer 2 Report
The review of previous works is poor, many references are too old, and many of them are not published in relevant journals Q1/Q2. A comparison with previous work published in relevant journals are necessary, highlighting the contribution of this paper. References should be over 30 papers.
Many grammar mistakes are presented along the manuscript, for example: “and has achieved a”, “curvature, and added”, “and obtained”, “improved.In” and many others.
I do not understand the meaning of “ ICP iteratively performed data association until convergence. ” in line 47.
Contribution is not well explained, please be clear with the novelty of this paper.
Hardware implementation and the comparison with other SLAM methods is very interesting, however the mathematical foundation of the proposed paper must be improved, discussion can be extended and better explained. Conclusions and future works are very poor and they are not interesting based on the implemented results presented in the real robot.
The explanation of the SLAM method proposed must be extended to a general mobile robot.
Author Response
- The review of previous works is poor, many references are too old, and many of them are not published in relevant journals Q1/Q2. A comparison with previous work published in relevant journals are necessary, highlighting the contribution of this paper. References should be over 30 papers.
Response: As suggested, We have added many recent reference papers and the number of reference papers has over 30, and highlight the contribution and the novelty in the introduction .
- Many grammar mistakes are presented along the manuscript, for example: “and has achieved a”, “curvature, and added”, “and obtained”, “In” and many others.
Response: As suggested, we check the full paper and make substantial improvements in English grammar.
- I do not understand the meaning of “ICP iteratively performed data association until convergence.” in line 47.
Response: As suggested, we check the full paper and make substantial improvements in English grammar. Specifically, we modify this sentence to “The ICP continues to repeat this process until convergence.”
- Contribution is not well explained, please be clear with the novelty of this paper.
Response: As suggested, we highlight the contribution and the novelty in the introduction.
- Hardware implementation and the comparison with other SLAM methods is very interesting, however the mathematical foundation of the proposed paper must be improved, discussion can be extended and better explained. Conclusions and future works are very poor and they are not interesting based on the implemented results presented in the real robot
Response: As suggested, we improved the mathematical foundations of the proposed method. And we extend the discussion. Besides, we revised the conclusion and future works.
- The explanation of the SLAM method proposed must be extended to a general mobile robot.
Response: The proposed method naturally depends on the sensors carried by the robot and has nothing to do with the specific robot type.
Round 2
Reviewer 1 Report
The author has answered the questions well. but still need some improvement. Writing and grammar must be improved. Here are some suggestions for the author:
1) On line 230, the author states that point clouds are clustered, but there is no explanation of how clustering is done. Give an explanation of the clustering carried out in this section.
2) Please add pseudocode to help readers understand this method better.
3) Please provide an explanation of the PC specifications used, materials used, and how many data or images are for testing.
4) In Section 6, provide references to the methods mentioned.
5) Tables 1 and 2 have the same title. Please check.
6) Please add discussion.
Author Response
First of all, all authors would like to thank you for your valuable comments that are very important to improve our paper. We carefully reviewed these comments and gave detailed responses as follows:
- On line 230, the author states that point clouds are clustered, but there is no explanation of how clustering is done. Give an explanation of the clustering carried out in this section.
Response: As suggested, we provide the clustering method used in the paper.
- Please add pseudo-code to help readers understand this method better.
Response: As suggested, we add the pseudo-code of the proposed method in Section 2.
- Please provide an explanation of the PC specifications used, materials used, and how many data or images are for testing.
Response: As suggested, we provide an explanation of the PC specifications used, sensor used, materials used, and the data used.
- In Section 6, provide references to the methods mentioned.
Response: As suggested, we provide the references to the methods mentioned in Section 6.
- Tables 1 and 2 have the same title. Please check.
Response: As suggested, we check and revise the table title.
- Please add discussion.
Response: As suggested, we add discussion and analysis in the conclusion section.
Reviewer 2 Report
The paper is improved compared with previous version. Conclusions can be presented with a more suitable analysis of the results.
Author Response
- The paper is improved compared with previous version. Conclusions can be presented with a more suitable analysis of the results.
Response: As suggested, we add discussion and analysis in the conclusion section.